# AI-Based Stroke Disease Prediction System Using Real-Time Electromyography Signals

**Jaehak Yu** [1], **Sejin Park** [1], **Soon-Hyun Kwon** [1], **Chee Meng Benjamin Ho** [1], **Cheol-Sig Pyo** [1] **and Hansung Lee** [2,*] 

[1] Department of KSB Convergence Research, Electronics and Telecommunications Research Institute (ETRI), Daejeon 34129, Korea; dbzzang@etri.re.kr (J.Y.); sjpark6840@etri.re.kr (S.P.); kwonshzzang@etri.re.kr (S.H.-K.); etri83341@etri.re.kr (C.M.B.H.); cspyo@etri.re.kr (C.-S.P.)

[2] School of Computer Engineering, Youngsan University, 288 Junam-Ro, Yangsan, Gyeongnam 50510, Korea

* Correspondence: mohan@ysu.ac.kr or mohan@korea.ac.kr

**Abstract:** Stroke is a leading cause of disabilities in adults and the elderly which can result in numerous social or economic difficulties. If left untreated, stroke can lead to death. In most cases, patients with stroke have been observed to have abnormal bio-signals (i.e., ECG). Therefore, if individuals are monitored and have their bio-signals measured and accurately assessed in real-time, they can receive appropriate treatment quickly. However, most diagnosis and prediction systems for stroke are image analysis tools such as CT or MRI, which are expensive and difficult to use for real-time diagnosis. In this paper, we developed a stroke prediction system that detects stroke using real-time bio-signals with artificial intelligence (AI). Both machine learning (Random Forest) and deep learning (Long Short-Term Memory) algorithms were used in our system. EMG (Electromyography) bio-signals were collected in real time from thighs and calves, after which the important features were extracted, and prediction models were developed based on everyday activities. Prediction accuracies of 90.38% for Random Forest and of 98.958% for LSTM were obtained for our proposed system. This system can be considered an alternative, low-cost, real-time diagnosis system that can obtain accurate stroke prediction and can potentially be used for other diseases such as heart disease.

**Keywords:** electromyography (EMG); stroke prediction; stroke disease analysis; artificial intelligence; machine learning; random forest; deep learning; long short-term memory (LSTM)

## 1. Introduction

The Fourth Industrial Revolution has arrived, bringing with it immense benefits as well as many challenges for various industries and research fields. In the healthcare field, the focus has been on information and communication technologies (ICT) such as artificial intelligence, big data, the Internet of Things (IoT), and cloud computing, which are also the core technologies for this Fourth Industrial Revolution [1–4]. For example, IoT enables the exchange of different types of health and medical information (bio-signals, past medical history, and genetic data) between various medical devices and medical institution systems. According to the WHO, the world's population is rapidly aging [5]; this will lead to increases in chronic diseases as well as healthcare costs. In anticipation of this, countries are shifting the focuses of their healthcare systems from sickness and disease to prevention and wellness. In addition, health information such as personal health records (PHR), electronic medical records (EMR), and genomic information is continually being generated, collected, and stored, and it can be easily used for analysis by making it big data. Over the years, a vast amount of medical data has been generated, collected, and stored, but it has yet to be properly used. An in-depth analysis of these data that combines big data and artificial intelligence (AI) technologies can help develop new

intelligent medical solutions, such as precision healthcare and predictive healthcare services, that can be used to prevent diseases. However, it is still difficult to derive meaning information between various types of healthcare big data. The recent improvements in computing infrastructure and the emergence of various AI frameworks in ICT have made AI-based digital healthcare analysis more intelligent and more feasible for this smart healthcare era. The smart healthcare system is evolving into a service that combines medical big data with ICT to help individuals manage their health remotely.

According to the World Health Organization (WHO), malignant neoplasms (cancer) and heart disease were the first- and second-leading causes of death in 2016, respectively. The third-leading cause of death in 2016 was stroke disease, accounting for 5.7 million deaths [5]. In 2016, stroke accounted for 11% of all deaths worldwide. A stroke occurs when there is an interruption of blood flow to the brain, resulting in necrosis of brain cells [6,7]. In general, stroke is more likely to occur in the elderly, and stroke can lead to cerebral dysfunction such as hemiplegia, mispronunciation, and lack of consciousness. These can cause severe disability in adults and even death [6,7]. Stroke is a treatable disease, and if detected or predicted early, its severity can be greatly reduced. Various studies and clinical trials have reported several risk factors of stroke [8–10]. Properly managing and treating adjustable risk factors such as hypertension, smoking, diabetes, and obesity can decrease stroke occurrences [10]. However, a survey of people with hypertension by the National Health and Nutrition Examination Survey (NHANES) found that about 30% of people were unaware that they had hypertension, 15% of patients did not take any medication, and 26% of hypertension cases were not yet well-controlled [11]. Therefore, medical and personal efforts are needed to improve this situation, and there is an urgent need to conduct research and implement national measures to prepare for the increase in other stroke diseases that is expected with the global aging trend.

It can be difficult to predict stroke symptoms or outbreaks using risk factors, because the definitions used for individual risk factors or the methodologies for correlation between the likelihood of a particular disease occurring are inconsistent. In particular, there are many issues with estimating the predictability of a precursor or the predictability of a stroke disease based only on the risk factors. The Framingham Heart Study, which was based on a forward-looking cohort study of cardiovascular disease, proposed a stroke risk prediction model [12,13]. However, a mismatch was observed with this model between the rates of heart disease predicted using a European population and those predicted using a Chinese population [13–16]. Therefore, there may be many differences in the socio-cultural environments and genetic characteristics of the subjects of the Framingham Heart Study. As a result, this limits the potential application of this model of previous research to Koreans and the elderly. In addition, stroke and the accompanying neurological damage represent a difficult domain to detect and predict early due to its various symptoms and classifications. Jee et al. [17,18] developed a stroke prediction model specifically for Koreans. This model was informed by 10 years of health examination data provided by the National Health Insurance Service (NHIS) [17]. However, like the stroke prediction model built by the Framingham Heart Study, it still has several disadvantages. These approach is based on past health examination data, i.e., risk factor. Since these provide the predicted probability of occurring a disease in the long distant future of about 5 or 10 years, it is difficult to apply to the various symptoms and predictions of the elderly (patient) before the onset of stroke at the present time.

This paper proposes a new stroke detection and prediction system based on real-time electromyography (EMG) data obtained from everyday activities. This system collected real-time EMG data during physical activities and used it for real-time prediction of stroke with the aid of Random Forest and Long Short-Term Memory (LSTM) algorithms. For improved prediction accuracy and validation of the system, the left and right biceps femoris and gastrocnemius muscle were measured and collected in real-time from a healthcare device at 1500 Hz. This system overcomes the limitations of previous studies in that it provides probability values for stroke disease in the next 10 years. In addition, it was experimentally verified that accurate detection and prediction of stroke disease is only possible with EMG bio-signals collected in real time. In this paper, our results showed stroke prediction accuracy

values of 90.38% for the Random Forest algorithm and of 98.958% for LSTM for participants while walking. This proof of concept indicates that a stroke can be detected using real-time bio-signals. The continuous monitoring of EMG ensures that the model is constantly updated, thus reducing misdiagnosis and allowing for quick responses from medical staff or hospitals. The prediction model of this proposed system can be used as a basic model for the pre-detection and the rapid prediction of other diseases such as heart disease.

This paper is organized as follows. Section 2 provides a background on the original research methods for stroke diseases. Section 3 introduces the artificial intelligence-based real-time stroke disease prediction model examined in this paper. Section 4 analyzes the experimental results and performance of the model, and finally, Section 5 concludes this paper and discusses future research tasks.

## 2. Related Works

### 2.1. Concept of Stroke

Stroke is one of the major diseases associated with death worldwide, and it causes cognitive and functional disorders [5,19,20]. A stroke happens when the blood vessels in the brain either become clogged or burst, thus reducing oxygen supply to the brain cells, resulting in necrosis of brain tissue [6,19]. As cells in the brain die, certain parts of the body lose functionality. This leads to various symptoms such as loss of body coordination, and speech and sensory impairment.

Stroke can be categorized into ischemic stroke (cerebral infarction), which is caused by blockages in the blood vessels, and cerebral hemorrhagic stroke (cerebral hemorrhage), which is caused by the rupturing of blood vessels in the brain [20]. Cerebral infarction diseases can be divided into cerebrovascular thrombosis and cerebral embolism. Cerebral thrombosis is a symptom that is caused by blocking blood clots in the brain due to arteriosclerosis or having problems with the inner wall of the blood vessel. While cerebral embolism is a condition caused by blood clots from the heart which blocks the blood vessels bring oxygen and blood to the brain. Next, there are two types of hemorrhagic stroke: intracerebral hemorrhage and subarachnoid hemorrhage. An intracerebral hemorrhage causes weak blood vessels to burst if there is a sudden rise in blood pressure. Brain cells that are supplied oxygen and nutrients by the arteries that have burst in turn become damaged, and the surrounding cells are crushed by the burst blood. Hypertension has been reported to be the main cause of most of these symptoms of intracerebral hemorrhage. Meanwhile, subarachnoid hemorrhage is caused by ruptured intracranial aneurysm irritating the lining of the brain. Subarachnoidal hemorrhage can be divided into spontaneous hemorrhage and traumatic hemorrhage. 80% of subarachnoid hemorrhage is caused by the ruptured cerebral aneurysm, and it is usually suspected when there is subarachnoid hemorrhage. Symptoms of subarachnoid hemorrhage vary from sudden severe headache, severe nausea, vomiting to loss of consciousness. However, the most characteristic symptom is a sudden severe headache unlike anything the person has experienced before. This subarachnoidal hemorrhage injury is reported to be fatal injury that cause death in one-third of patients before they can arrive at a hospital, and only the remaining patients are known to receive treatment [19,20].

According to Statistics Korea, the total number of deaths in Korea in 2018 was 298,820, with 161,187 for men and 137,633 for women [21]. The causes of death were reported to be 79,153 cases of malignant neoplasm (cancer), 32,004 cases of heart disease, 23,280 cases of pneumonia, and 22,940 cases of cerebrovascular disease. Cerebrovascular disease is the third-highest single disease cause of death, with high mortality rates of 42.7 for men and 46.7 for women per 100,000 [21]. Although the death toll from cerebrovascular disease has been declining since 2005, it is still a high-risk disease that ranks third among all single disease causes of death. In particular, for those aged 60 or older, the mortality rate from heart disease and cerebrovascular disease is gradually increasing. Fast detection and treatment are paramount in the early onset of a stroke, as untreated stroke can leave severe aftereffects, such as hemiplegia or even death.

## 2.2. Stroke Prediction Using Traditional Techniques

Successful research has been conducted to monitor the conditions of stroke patients and discover major risk factors for stroke. This knowledge can be used to prevent the recurrence of stroke and assess patient stroke severity [22,23]. As methods of assessing stroke patients' severity, the European Stroke Scale, the Canadian Neurologic Scale, and the National Institutes of Health Stroke Scale (NIHSS) have been published since the Mathew scale was first published in 1972 [22,23]. Of these, the NIHSS is widely used around the world and has been proven in terms of reliability and validity. The NIHSS scale consists of 14 items: level of consciousness; examination; vision; facial paralysis; upper and lower extremities; impaired limbs, senses, and speech; oral disorders; neglect; and distal movement. It takes about 6.6 min for a patient to be measured on the NIHSS. A modified NIHSS [24] that quantitatively evaluates post-stroke disorders and simplifies the measurement process, particularly in the early stages of hospitalization. While these NIHSSs can comprehensively assess the severity of disability resulting from stroke, they cannot be used for the early detection of stroke, as they do not provide accurate predictions.

A stroke prediction model for Koreans was developed by Jee et al. [25]; this model can determine the risk of stroke occurrence for 10 years based on health examination data from the National Health Insurance Service (NHIS), including age, diabetes, hypertension, smoking, total cholesterol, exercise, body mass index, and drinking volume. However, that study has limitations similar to those of the Framingham heart study model, as the same method was used to create the stroke risk prediction model [12,13]. The main issue with these models is that they do not take into account some major risk factors of stroke and death causes other than stroke, i.e., competing risk. From the onset of a stroke, patients must receive professional treatment within three hours. In this three-hour period, the type of stroke (ischemic stroke or hemorrhagic stroke) and the severity of the stroke need to be determined to administer the right treatment [12,13,26]. Therefore, in order to accurately detect and predict these factors during daily life, it is necessary to not only apply PHR and EMR, but also to analyze information regarding patterns and bio-signals in daily life from a healthcare device.

In other studies, major risk factors for stroke were identified through prior studies and clinical trials, and these risk factors were reported to include smoking, hypertension, systolic blood pressure, diabetes, and obesity [12,13]. In other words, stroke is more likely to be caused by the interaction of various risk factors rather than any one factor. Thus, a new methodology is emerging that can assess the risk factors of each individual to predict stroke early. Based on these risk factors, research was conducted with various statistical methods such as a logistic model [27], Cox's proportional risk model [12,13], and the Weibull model [28]. However, these risk-based advanced studies are not appropriate for predicting the risk of stroke in Koreans. There is a particular need to find a new model that can be used to predict stroke risk for Korean elderly people. The onset of a stroke can be sudden, and it is reported to be nine times more likely for a stroke to reoccur than it is to occur in those with no history of stroke [29]. In addition, clinical studies have reported that stroke recurrence rates vary depending on the type of stroke, the race of the person, and various risk factors, but within one year, stroke generally has a recurrence rate of 10–15%. It is therefore important to quickly detect and predict the initial outbreak of stroke as well as possible recurrence in people with prior stroke history [30].

Clinical studies have indicated that initial changes in physiological variables are potential therapeutic targets for stroke. The correlation between a patient's physiological variables 48 h after the stroke and the result of stroke severity after three months was reported by Zhang et al. [31]. That team used regression techniques to build algorithms that could predict the results of three months in early physiological data and showed 71% test accuracy with the statistical characteristics of physiological data alone. In particular, it was reported that the trend pattern of physiological time series data had an important meaning in the initial treatment of acute ischemic stroke patients [31]. Chien et al. [32] performed a cohort study of adults in Taiwan regarding the risk of stroke in 10 years. That study proposed a prediction method using the Cox model based on clinical and biochemical models as well as net screening and integrated screening assessment statistics. That model gives important weight to

each attribute based on patient-specific health examinations and historical data. The study revealed various major stroke prediction factors, such as age, gender, systolic blood pressure, relaxation blood pressure, family stroke status, atrial fibrillation, and diabetes. Song et al. [33] showed the link between ischemic and hemorrhagic stroke incidence during sleep in the general public. Sleep was divided into three classes: less than six hours of sleep, six to eight hours of sleep, and eight hours of sleep or more; ischemic and hemorrhagic strokes were then observed for two years. According to the analysis results, women over the age of 65 reported a higher incidence of hemorrhagic stroke during longer sleep periods of more than eight hours.

### 2.3. Stroke Prediction Using Machine Learning and Deep Learning

Several studies have used artificial neural networks (ANN) for stroke diagnosis or prediction [34–37]. Based on stroke patient data, Shanthi et al. [36] reported that the risk of stroke can be detected with ANN. In that study, the backpropagation algorithm was used for learning, and the consistency of prediction and diagnostic accuracy were improved. Based on ANN, Kasabov et al. [35] were able to predict the risk rate of stroke using 300 experimental data. As a result of the experiment, a model that can match the data of stroke patients with a high accuracy of 95.33% was proposed. However, that system only emphasizes prediction accuracy, and it is difficult to analyze the mechanics of the operating principle. Bentley et al. [38] examined symptomatic intracranial hemorrhage during melting treatment for ischemic stroke, and reported a predictive method considering data from CT (Computerized Tomography) images and clinical variable values. Specifically, based on CT images of 116 ischemic stroke patients, nine out of 16 patients with symptomatic intracranial hemorrhage were successfully identified using SVMs. Khosla et al. [39] reported the prediction and verification results by adjusting the parameter values of the predictive model using the kernel function of SVM for the risk factors of stroke. In particular, the RBF kernel function was used to obtain satisfactory accuracy, and this model details the process of predicting stroke risk. However, these SVM-based studies focused on predicting severity and prognosis after an outbreak, as opposed to early detection or the prediction of pre-occurrence symptoms. Further, it is hard to explain this model from an analytical point of view, as the system's operating principles are a black box; this is because it only maintains the position of increasing the accuracy of traditional stroke predictions. In addition, since it is based on images of separate clinical diagnosis and CT, it is not suitable for detecting and predicting pre-symptoms of stroke diseases using real-time bio-signals or life logs during daily life. Therefore, new research and trials are needed for the early detection of stroke onset.

To overcome the shortcomings of the NIHSS studies described in Section 2.2, Yu et al. [18] published an analytical study using data mining techniques. Yu et al. performed an analytical study based on the decision tree algorithm, a representative classification methodology of data mining; specifically, they attempted to automatically classify and interpret the severity of NIHSS results based on the C4.5 decision tree algorithm. In addition, by thoroughly analyzing the rules on the principle of motion, which provide additional information for the C4.5 decision tree, a very novel attempt was made in the semantic interpretation of stroke severity. However, due to the nature of the decision tree algorithm, the predictive model algorithm only provides a partial interpretation. and may require detailed analysis inherent in the data. Another study was conducted by Amini et al. [40] to predict stroke based on more than 50 risk factors for stroke. In that paper, the prediction experiment results of relatively accurate stroke are presented with algorithms such as KNN (K-nearest neighbors) and the C4.5 deception tree for data mining. However, this research methodology, like those used in previous studies, is not suitable for use as a prediction model for detecting and predicting the pre-symptoms of stroke in everyday life.

Recurrent Neural Network (RNN) is a representative method of analyzing and processing time-series data in deep learning. RNNs are composed of circulatory neural networks, which are used in the current learning process along with the results of previous steps [41]. In an RNN structure, the current output results of cells in the neural network are affected by the previous calculation results,

and it has an advantage in learning sequential data because it has memory information about the previous calculation results. Long Short-Term Memory (LSTM), a type of RNN, is a model that overcomes the structural shortcomings of the existing RNN and can solve the problem of calculating and decreasing values when error values are propagated to the neural network layer. This LSTM was first proposed by Hochreiter and Schmidhuber in 1997 [42]. Bloomberg Business Week in the United States described LSTM as AI that can be used and activated in various fields, including disease prediction and music composition. These LSTMs consist of a cell state, an input gate, a forget gate, and an output gate, and vector output values are generated at each gate through the sigmoid layer and the tanh layer. The cells of LSTM learn to recognize important inputs (input gate), learn to preserve them for a long period of state storage and defined time, and execute learning to extract them whenever necessary [43,44]. In a recent research trend related to LSTMs, studies have attempted to analyze the risk factors of EHRs (Electronic Healthcare Records) to predict LSTM-based cerebrovascular diseases [45]. Specifically, by incorporating ICD-10 codes [46] and other potential risk factors patterns from EHRs, Chantamit [45] confirmed that the LSTM algorithm is the most suitable for predictive analysis of any cerebrovascular disease or stroke. In another study, a methodology based on the LSTM model for predicting HDM (hemorrhagic transformation) in ischemic stroke was proposed by Yu et al. [47]. The LSTM network structure was designed using a combination of DWI (diffusion weighted images) and PWI (perfusion-weighted magnetic response images). A comparative analysis of 155 acute stroke patients collected from clinical trials showed 89.4 percent accuracy. While these studies have shown the potential of LSTM for the prediction of stroke, they have still used data such as EHR and images. In other words, there has been no research on the prediction and analysis of stroke using real-time bio-signals that are generated from everyday activities, such as walking and driving. Therefore, a new methodology for predicting stroke using real-time bio-signals is required as an alternative to the existing traditional methodology.

## 3. Artificial Intelligence-Based Stroke Disease Prediction System Using EMG

In this paper, we propose a new AI-based stroke disease system using EMG bio-signals from everyday life as shown in Figure 1. The first module executes offline processing and includes the functions of machine learning and the deep learning-based learning model generation and management of EMG data. The second module performs an online processing function as well as early detection and prediction of stroke based on EMG bio-signals collected in real-time from everyday life. In the offline module, the EMG bio-signals data measured in real-time during daily walking is updated in the repository according to the cycle set by the system. Preprocessing of the collected EMG bio-signals is performed, and a learning model is generated using machine learning and deep learning algorithms. Attribute subset selection from the preprocessed EMG is used to generate learning models with machine learning algorithms for early detection and prediction of stroke diseases. Machine learning LSTM-based learning models are developed and sent for online processing to the second module before being used in real-time stroke predictions. The online module measures and collects real-time EMG bio-signals in daily life at the request of the system or user. The "preprocessing and normalization" block removes missing or incomplete data from the collected EMG data. As the minimum and the maximum values are different for each attribute, a normalization process is performed depending on the measurement unit. In the attribute subset selection block, the prediction accuracy and the analysis speed are improved by selecting the EMG attribute subset defined in this paper. In addition, it is possible to guarantee optimal performance of the predicted model learned and provide analytical information. The 'Real-Time Stroke Prediction' block is mounted with pre-trained machine learning and LSTM-based prediction models. Real-time prediction and semantic analysis based on machine learning are performed using the selected EMG optimal attributes subset. At this time, the deep-learning LSTM model implements early detection and prediction of stroke in real-time based on EMG bio-signals that went through preprocessing blocks. These predictions and semantic analysis information are then transmitted to the medical staff or hospital. Finally, based on the medical doctor's diagnosis for the

risk of stroke, the patients are provided assistance with receiving medical examinations and treatment services with emergency alarms and quick hospital visits as appropriate.

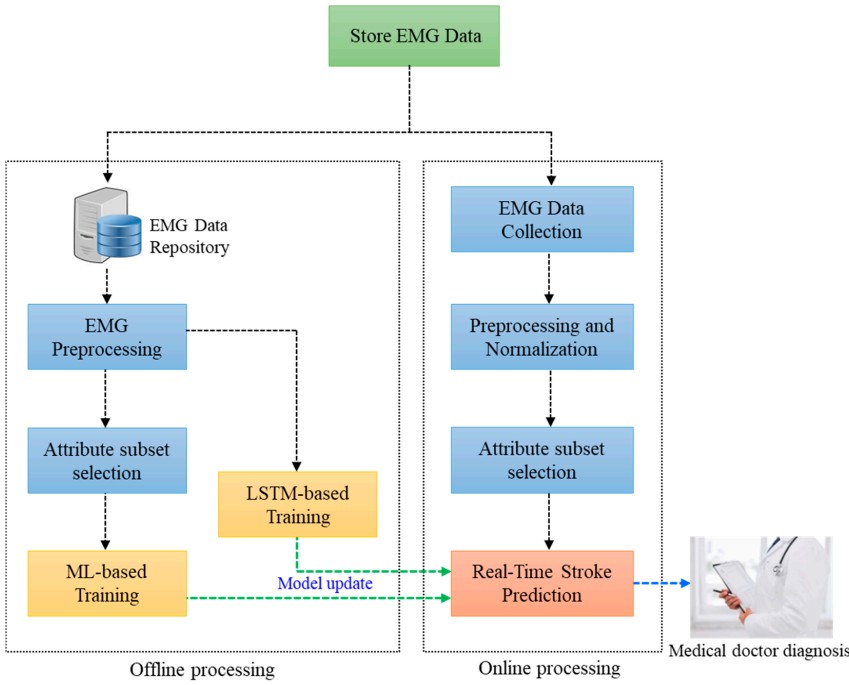

**Figure 1.** Overall structural diagram of AI-based stroke disease prediction system using EMG bio-signals.

## 4. Experiments and Analysis

### 4.1. Dataset and Experimental Analysis

This section describes the process of measuring and collecting bio-signal data for verification of the proposed AI-based stroke disease prediction system. The main bio-signal used will be real-time electromyography (EMG) data. EMG is a diagnostic tool that records electrical activity inside a specific muscle or measures nerve conduction velocity through electrical stimulation using electrodes. Comprehensive studies of strokes using EMG have shown that there is a slight imbalance in the body both before and after a stroke, along with imbalances in gait and locomotion [48,49]. In this paper, we analyze these imbalances and gait disorders in terms of EMG bio-signal data, which has the potential to be a stroke risk factor.

The measurement and collection of EMG biometric data were conducted from 2015 to 2017 at the emergency medical center and the department of rehabilitation medicine at Chungnam National University Hospital. Our stroke group consisted of patients aged 65 or older who were undergoing rehabilitation treatment for stroke within one month of stroke confirmation. In total, 287 patients from the rehabilitation department fit our criteria. Various biological signals were collected, such as EMG, ECG (electrocardiogram), EEG (electroencephalogram), foot pressure, and voice recordings. Figure 2 shows the location of each sensor on the patient. Before collecting bio-signal data through experimentation, the sensors were checked to confirm they were operating normally. Patients who were undergoing rehabilitation other than a stroke were considered as the normal group, and data were collected from 271 individuals. Each patient was put through various scenarios such as standing, walking, sitting, raising arms, and sleeping to simulate everyday activities. For each scenario, all subjects had one practice before executing the measurement protocol. Despite this prior practice, the first measured and collected values were not used as experimental data, because human noise due to the subject's tension and discomfort could be reflected. At the same time, the last measurement protocol was also not reflected because repetitive experiments cause fatigue to elderly subjects. Each bio-signals

data collection information was delivered to the relay in real time and the BLE (bluetooth low energy) protocol was used for communication. In addition, data was forwarded to the Wi-Fi communication protocol to the server that collects and predicts bio-signals from the gateways. At this time, the entire process of the measurement experiment was monitored by the medical doctor and re-measurement was performed if there was a loss of bio-signal data transmitted. The raw data transmitted from the four locations of the EMG was set in the system to have a false value of 4 bytes per sampling rate in the form of a voltage.

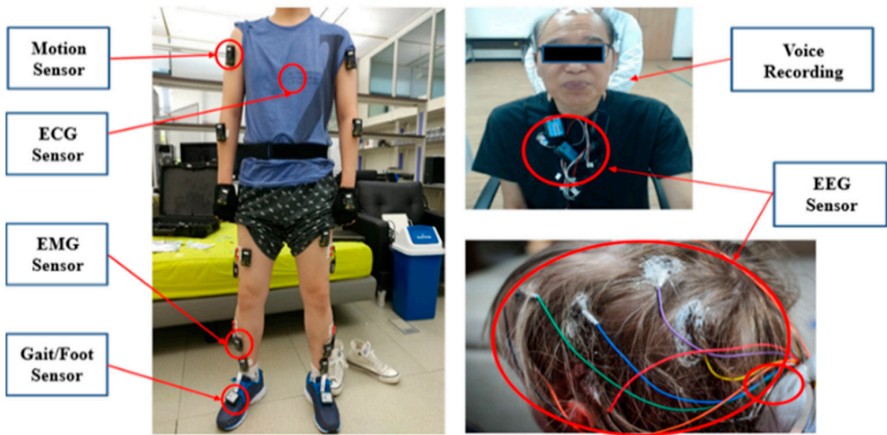

**Figure 2.** Sensor specific location attachment for collecting multiple bio-signals.

Figure 3 shows the measurement locations used to collect the EMG bio-signals. EMG signals were collected with a sampling rate of 1500 Hz per second from a total of four locations: the left and right legs, biceps femoris, and gastrocnemius muscle. As a stroke patient and a normal person should be classified based on LSTM of machine learning and a deep learning model, it is necessary to ensure an equal amount of experimental data for the two classes for predicting a stroke. Therefore, 271 out of 287 stroke patients were randomly extracted, and 271 normal subjects' data were used.

Figure 4 shows raw EMG signals for a random stroke patient and a normal patient collected at the four muscle locations while walking, which will be used in the model training and the testing of LSTM.

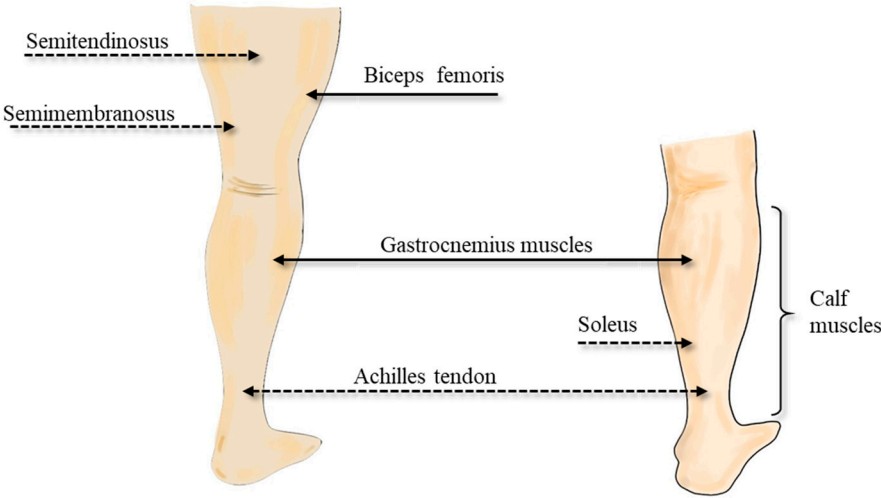

**Figure 3.** Real-time measurement location for EMG bio-signal (Left image: biceps femoris, right image: gastrocnemius muscle).

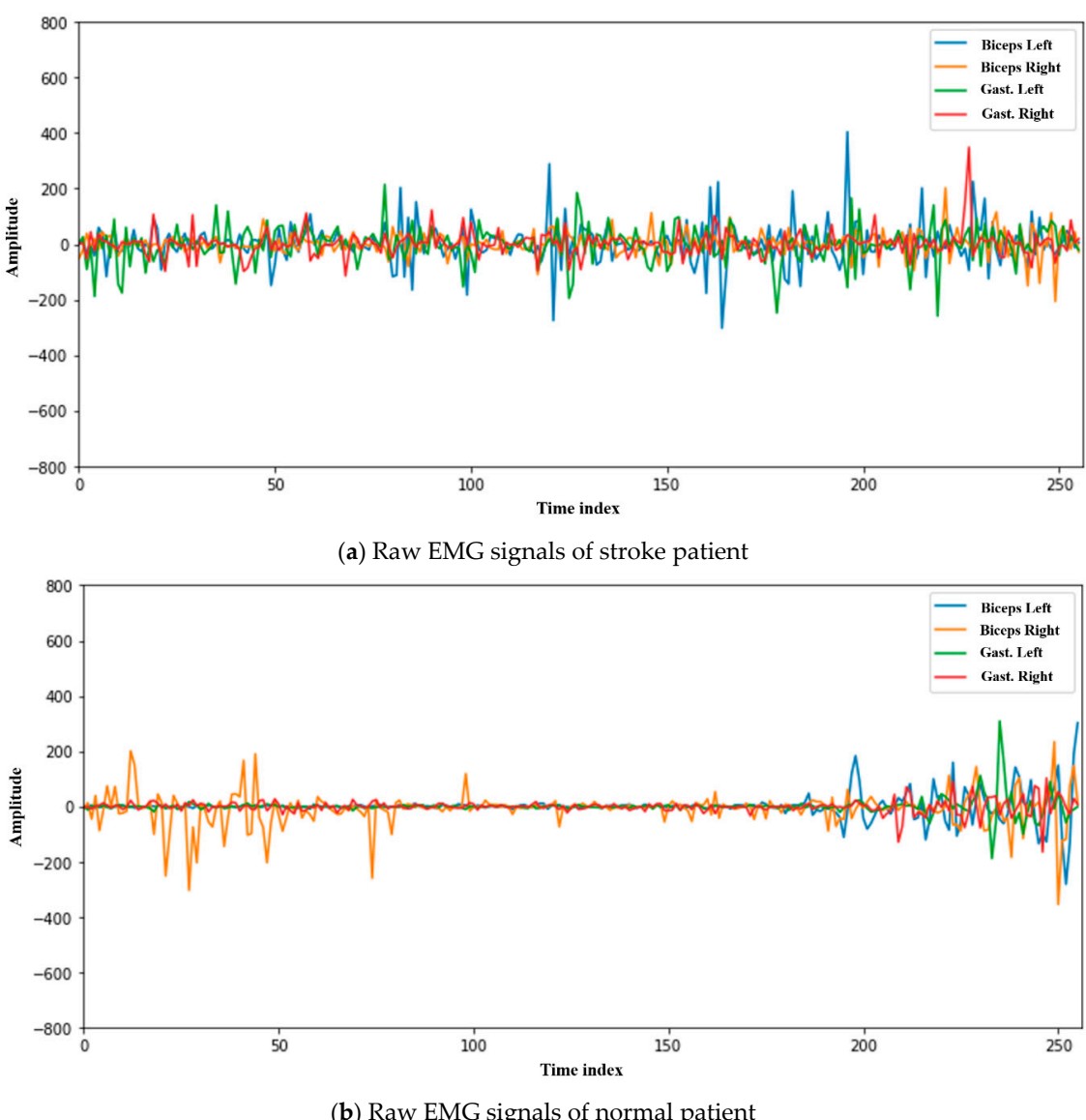

(**a**) Raw EMG signals of stroke patient

(**b**) Raw EMG signals of normal patient

**Figure 4.** Example of EMG bio-signals waveform changes while walking. (**a**) stroke patient sample, (**b**) normal sample.

## 4.2. Experiment and Analysis Based on Machine Learning

EMG bio-signals measure muscle response or electrical activity in response to a nerve's simulation, and they are used to diagnose neuromuscular disorders and balance abnormalities. In this paper, 28 attributes are newly defined and extracted to predict stroke disease based on machine learning using EMG bio-signals. The attributes are extracted from the raw data of the left and right biceps femoris and gastrocnemius muscles (see Table 1). The attributes extracted from EMG raw data were used for the machine learning-based prediction model experiments and multidimensional analysis. At 1500 Hz per second EMG, the variation in muscle movement is considered to be acceptable, and data points were extracted by dividing the EMG raw data into 0.1 s units when extracting the attributes [50–53]. Data measurements were collected based on the protocols defined in this paper. Our scenario includes standing, walking, raising arms, and sleeping, and only EMG bio-signals collected at this time were used as experimental data. Although experimental data were collected for various activities, only walking data was considered. Table 1 below presents detailed descriptions of the newly defined and extracted EMG attributes in this paper.

**Table 1.** Detailed Descriptions of Newly Defined and Extracted EMG Attributes.

| Number / Contents | Attributes | Description |
|---|---|---|
| 1 | BFL_Mmax | The value from the lowest point of the negative peak of the left biceps femoris to the positive peak. |
| 2 | BFR_Mmax | The value from the lowest point of the negative peak of the right biceps femoris to the positive peak. |
| 3 | LGL_Mmax | The value from the lowest point of the negative peak of the left gastrocnemius muscle to the positive peak. |
| 4 | LGR_Mmax | The value from the lowest point of the negative peak of the right gastrocnemius muscle to the positive peak. |
| 5~8 | BFL_PositivePeak, BFR_PositivePeak, LGL_PositivePeak, LGR_PositivePeak | The maximum value of the positive peak based on 0 of biceps femoris/gastrocnemius muscle on the left and right. |
| 9~12 | BFL_NegativePeak, BFR_NegativePeak, LGL_NegativePeak, LGR_NegativePeak | The minimum value of the negative peak from 0 of the left and right femoral/gastrocnemius muscles. |
| 13~16 | BFL_5_PPIMean, BFR_5_PPIMean, LGL_5_PPIMean, LGR_5 PPIMean | The average value of the five-forward positive peak-interval (PP-I) based on the current positive peak point of the left/right biceps femoris/gastrocnemius muscle. |
| 17~20 | BFL_10_PPIMean, BFR_10_PPIMean, LGL_10_PPIMean, LGR_10 PPIMean | The average value of the 10-forward PP-I based on the current positive peak point of the left/right biceps femoris/gastrocnemius muscle. |
| 21~24 | BFL_5 PPISD, BFR_5 PPISD, LGL_5 PPISD, LGR_5 PPISD | The standard deviation of the five-forward PP-I based on the current positive peak point of the left/right biceps femoris/gastrocnemius muscle. |
| 25~28 | BFL_10 PPISD, BFR_10 PPISD, LGL_10 PPISD, LGR_10 PPISD | The standard deviation of the 10-forward PP-I based on the current positive peak point of the left/right biceps femoris/gastrocnemius muscle. |
| 29 | Class Labeling | Normal or Stroke |

In the first experiment, 271 stroke patients and 271 normal patient subjects were tested. The measurement protocol used all 28 attributes obtained from the EMG signals during walking. Table 2 presents the accuracy of each machine learning model based on the EMG data from a total of 542 people. The data set configuration for the experiment was tested by randomly selecting data for training and testing at ratios of 70 to 30 and 80 to 20, respectively, based on the total data. Further, the reliability of the models may be reduced if the number of data is small in a single experiment. Thus, K-fold cross validation was used to compensate for these shortcomings and to analyze how well the dependent variable values in the test data were predicted. As a result of the experiment and analysis, the Random Forest algorithm using all 28 attributes confirmed the highest prediction accuracy of 85.82% in 20-fold cross validation (CV).

**Table 2.** Prediction accuracy of stroke by machine learning algorithm with all 28 attributes (%).

| Methods / Data Sets | Train (70)/ Test (30) | Train (80)/ Test (20) | 5-Fold CV | 10-Fold CV [1] | 20-Fold CV |
|---|---|---|---|---|---|
| C4.5 Decision Tree | 78.22 | 78.23 | 78.78 | 79.65 | 79.43 |
| C5.0 Decision Tree | 79.48 | 79.68 | 80.01 | 80.32 | 80.29 |
| Naïve Bayes | 62.82 | 63.06 | 62.81 | 62.80 | 62.85 |
| Logistic Regression (LR) | 71.45 | 70.25 | 70.33 | 70.33 | 70.33 |
| ANN (MLP) | 66.16 | 59.17 | 64.47 | 68.63 | 66.15 |
| Random Forest (RF) | 85.03 | 85.35 | 85.67 | 85.78 | 85.82 |
| C&RT | 77.25 | 77.38 | 77.44 | 77.48 | 77.59 |
| CHAID | 77.01 | 77.12 | 77.57 | 77.41 | 77.37 |
| Two-Class SVM | 71.12 | 70.58 | 71.18 | 71.47 | 71.51 |

[1] CV: Cross Validation.

In the second experiment, stroke prediction performance was verified by applying quartiles to the 28 attributes. Medical data (EHR) are typically eliminated using a statistical method of quartiles for missing and out-of-range values. In this experiment, the stroke disease prediction was applied to remove the corresponding data rows by quartile, and the experimental results presented in Table 3 below were obtained. As with the first experiment, the Random Forest algorithm showed the best performance, with predicted accuracies of 85.86% and 85.84% at 10-fold CV and 20-fold CV, respectively.

**Table 3.** Prediction accuracy of stroke after quartile application (%).

| Data Sets / Methods | Train (70)/ Test (30) | Train (80)/ Test (20) | 5-Fold CV | 10-Fold CV | 20-Fold CV |
|---|---|---|---|---|---|
| C4.5 Decision Tree | 81.25 | 81.42 | 81.73 | 82.46 | 81.97 |
| C5.0 Decision Tree | 81.68 | 81.78 | 82.44 | 82.65 | 82.61 |
| Naïve Bayes | 69.60 | 69.54 | 69.88 | 69.89 | 69.86 |
| Logistic Regression (LR) | 71.16 | 71.09 | 71.23 | 71.26 | 71.28 |
| ANN (MLP) | 77.98 | 77.94 | 78.56 | 78.78 | 78.79 |
| Random Forest (RF) | 85.38 | 85.44 | 85.70 | 85.86 | 85.84 |
| C&RT | 79.89 | 79.86 | 79.95 | 80.16 | 80.12 |
| CHAID | 79.08 | 79.03 | 79.38 | 79.59 | 79.58 |
| Two-Class SVM | 73.33 | 73.31 | 73.63 | 74.07 | 73.72 |

The third experiment compared the accuracy of the prediction model with data that applied the method of selecting quartiles and correlation-based feature selection (CFS) on the original data (see Table 4). CFS is a method of reducing the size of a data set by eliminating duplicate or unrelated features for classification. In addition, for data classification that involves finding the optimal set of features suitable for the goal, the probability distribution will be similar to or even higher than the probability distribution obtained using all the features. For this experiment, not all of the extracted features were used; instead, the Hall [54] method was used to select a subset of features. Hall's method calculates conditional probabilities using entropy for the attribute value and the best first search value, $Y$, as well as Pearson's correlation coefficient between the target class and the attributes. First, the entropy was calculated for any attribute $Y$ as shown in Equation (1) to obtain information benefits for each attribute [54,55].

$$H(Y) = -\sum_{y \in Y} p(y) log_2(p(y)) \tag{1}$$

**Table 4.** Prediction accuracy of stroke with quartile and CFS (%).

| Data Sets / Methods | Train (70)/ Test (30) | Train (80)/ Test (20) | 5-Fold CV | 10-Fold CV | 20-Fold CV |
|---|---|---|---|---|---|
| C4.5 Decision Tree | 80.08 | 80.58 | 81.90 | 82.20 | 82.27 |
| C5.0 Decision Tree | 82.48 | 82.65 | 83.11 | 83.15 | 83.18 |
| Naïve Bayes | 69.25 | 69.14 | 69.58 | 69.60 | 69.58 |
| Logistic Regression (LR) | 71.08 | 70.90 | 71.13 | 71.16 | 71.13 |
| ANN (MLP) | 76.70 | 77.73 | 77.62 | 77.53 | 77.68 |
| Random Forest (RF) | 85.94 | 85.88 | 86.52 | 86.89 | 86.88 |
| C&RT | 80.59 | 80.56 | 80.72 | 80.89 | 80.78 |
| CHAID | 80.42 | 80.37 | 80.67 | 80.91 | 80.98 |
| Two-Class SVM | 74.01 | 73.98 | 74.24 | 74.67 | 74.15 |

Next, the merit function (Equation (2)) was used to assess how efficiently each subset $F_s \subset F$ expresses the overall attributes. The value of the merit function means that the subset with the largest value is the subset that best represents the entire attributes set [54].

$$Merit(F_s) = \frac{k\overline{r_{cf}}}{\sqrt{k + k(k-1)\overline{r_{ff}}}} \tag{2}$$

where $k$ is the number of attributes in subset $F_s$, $\overline{r_{cf}}$ is the mean distribution of attributes contained in $F_s$, and $\overline{r_{ff}}$ is the mean correlation value of the attributes. The 20 attributes for the subset that were finally selected through CFS are as follows, listed in order of importance: BFL_PositivePeak, BFL_NegativePeak, BFL_5 PPISD, BFL_10 PPISD, BFR_Mmax, BFR_PositivePeak, BFR_NegativePeak, LGL_Mmax, LGL_PositivePeak, LGL_NegativePeak, LGL_5_PPIMean, LGL_10_PPIMean, LGL_5 PPISD, LGL_10 PPISD, LGR_Mmax, LGR_PositivePeak, LGR_NegativePeak, LGR_5 PPIMean, LGR_10 PPIMean, and LGR_10 PPISD. The prediction accuracy of stroke when using only 20 attributes with CFS is improved by more than 1% when compared to the prediction using all 28 attributes. For disease prediction, an accuracy improvement of more than 1% has great significance. We showed that by using the CFS method, only key attributes need to be selected and utilized, and the prediction accuracy can still be improved. This reduces the time needed for classification and maximizes the efficient use of system resources.

In the fourth experiment, 20 attributes were finally selected by applying quartiles and attribute subsets (CFS) to the original data, and then Z-score normalization was performed for each attribute as shown in Equation (3) (see Table 5). This normalization process converts the data to a small range, such as from 0.0 to 1.0, so that the same weight can be applied to all attributes. For example, the minimum and the maximum values of the BFL_PositivePeak vary in size and are subject to measurement units, and therefore need to be avoided. In Equation (3), $\sigma$ and $\mu$ are the standard deviation and mean of attribute $x$, respectively, and $\alpha$ is set as the weighted value, which was 1.0 in the experiment.

$$\vec{x_i} = \frac{x_i - \mu}{\sigma} \times \alpha \tag{3}$$

**Table 5.** Prediction of stroke by machine learning algorithm after normalization (%).

| Data Sets / Methods | Train (70)/ Test (30) | Train (80)/ Test (20) | 5-Fold CV | 10-Fold CV | 20-Fold CV |
|---|---|---|---|---|---|
| C4.5 Decision Tree | 84.23 | 84.94 | 83.89 | 84.39 | 84.56 |
| C5.0 Decision Tree | 84.41 | 84.58 | 84.86 | 85.02 | 85.16 |
| Naïve Bayes | 68.37 | 68.44 | 68.89 | 68.91 | 68.89 |
| Logistic Regression (LR) | 70.04 | 70.23 | 70.33 | 70.28 | 70.31 |
| ANN (MLP) | 75.60 | 75.58 | 76.01 | 75.98 | 75.96 |
| Random Forest (RF) | 89.42 | 89.66 | 89.92 | 90.25 | 90.38 |
| C&RT | 81.04 | 81.07 | 81.81 | 81.75 | 81.73 |
| CHAID | 80.98 | 80.93 | 81.02 | 81.11 | 81.15 |
| Two-Class SVM | 74.55 | 74.62 | 74.55 | 74.81 | 74.83 |

The result showed that the Random Forest algorithm had high prediction accuracies of 90.25% and 90.38% at 10-fold CV and 20-fold CV, respectively. Since the Random Forest algorithm is an ensemble method that involves learning from several decision trees, it overcame the disadvantage of a large variation in the performance of the decision tree and ultimately showed good accuracy. Furthermore, some machine learning methodology only places importance on the prediction accuracy of the disease and may overlook the machine interpretation of the operating principles of the system. These core operating principles are like black boxes that allow for limited semantic analysis based on medical or clinical data. For this reason, it is a desirable approach to make predictions through a Random

Forest algorithm based on a decision tree that has a heuristic methodology but also has analytical and analytical advantages.

### 4.3. Experiment and Analysis Based on Deep Learning

This section will go through the results of LSTM which are based on a Recent Neural Network (RNN) for the classification of stroke patients and non-stroke patients in the elderly. Raw data from EMG bio-signals were used as defined in Section 4.1. Data from 271 stroke patients and 271 normal patients were used to generate prediction models and verify their accuracies. Learning data sets were randomly extracted, and the data not used for learning became the test sets. The experiment and analysis were carried out by dividing them into ratios of 70 to 30 and 80 to 20, respectively. LSTM was chosen among other deep learning algorithms due to its performance in the time series analysis. This LSTM consists of a structure that explicitly transfers past information to the next state, and by learning long-term dependencies, the cell state serves to convey past information to the next step [41–44]. Deep Learning's LSTM has overcome the structural shortcomings of the existing RNN and can solve the problem of having a vanishing gradient when error values are propagated to the neural network layer [41,42,45,47]. LSTM consists of a cell state, an input gate, a forget gate, and an output gate, and vector output values at each gate are generated via the sigmoid layer and the tanh layer (see Figure 5).

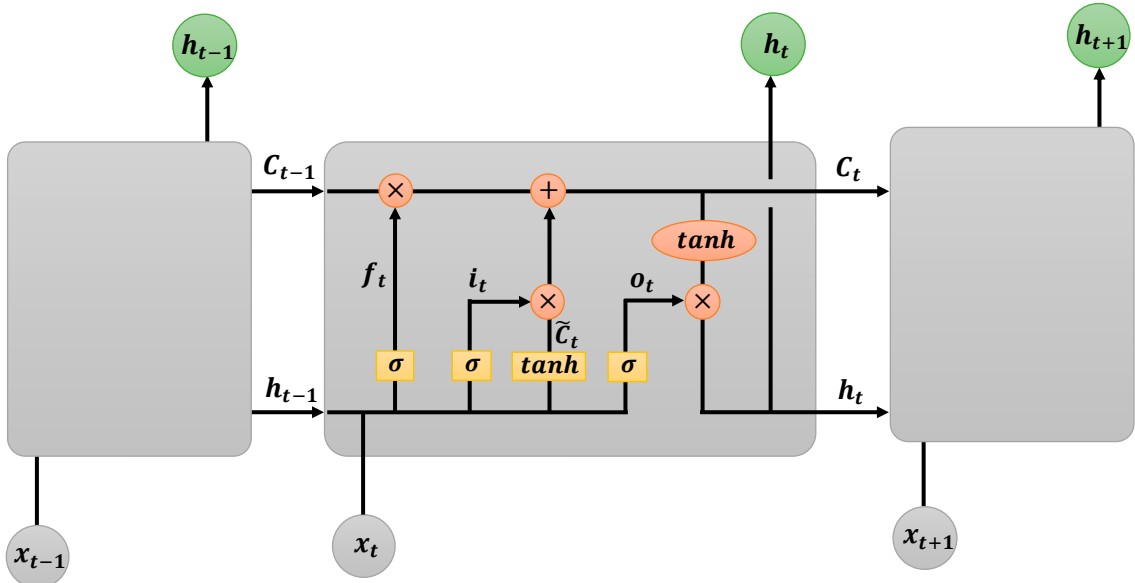

**Figure 5.** Structural Diagram of LSTM based on EMG Biometric Signals.

Before the experiment, the Z-score normalization process was carried out as shown in Equation (3) of Section 4.2 for each point of raw EMG bio-signals data from the left and right biceps femoris and gastrocnemius muscle. In this experiment, the same weights were applied for each measurement value by converting the values of the four raw data by location into smaller ranges from 0.0 to 1.0. The $x_t$ refers to the input vector value for the four bio-signals of the EMGs at point $t$. The forget gate layer takes $h_{t-1}$ and $x_t$ as input values. The input $x_t$ outputs $f_t$, which is the value given by the activation function sigmoid function $\sigma$, and determines whether or not the information is transferred to the cell state as shown in Equation (4) below.

$$f_t = \sigma\left(W_f \cdot [h_{t-1}, \; x_t] + b_f\right) \tag{4}$$

Since $f_t$ is the output value of the sigmoid function, it has a value between 0 and 1. The closer this value is to 1, the more likely it is that the output will be kept. In other words, the fact that $f_t$ is close to 1 means that from the cell state's point of view, it is influenced by the long-term memory of the past. This can be interpreted as maintaining the gradient for a long-term depending on the time step. The input gate determines the information to be updated in the cell state and is shown in Equation (5) below. The layer then takes the activation function hyperbolic tangent to generate the value $\widetilde{C}_t$ to add to the cell state (see Equation (6)).

$$i_t = \sigma\left(W_i \cdot [h_{t-1},\ x_t] + b_f\right) \tag{5}$$

$$\widetilde{C}_t = tanh(W_C \cdot [h_{t-1},\ x_t] + b_C) \tag{6}$$

Equation (7) generates a new cell state $C_t$. $\widetilde{C}_{t-1}$ is updated by multiplying the old state by $f_t$, then adding $i_t$ and $\widetilde{C}_t$. After performing the element-wise product, a new cell $C_t$ is created, which will be delivered to the next time step.

$$C_t = f_t * C_{t-1} + i_t * \widetilde{C}_t \tag{7}$$

Equations (8) and (9) show the role of the output gate, the output value of the sigmoid layer $o_t$, and the updated cell state value $C_t$. The element-wise product is run to generate the output value of $h_t$, which is the value taken by the hyperbolic tangent function on $C_t$.

$$o_t = \sigma(W_o \cdot [h_{t-1},\ x_t] + b_o) \tag{8}$$

$$h_t = o_t * tanh(C_t) \tag{9}$$

This paragraph describes important items and parameters for the experiments and verification using the EMG bio-signals-based LSTM model. Iteration means the maximum number of times for learning, and nUnit indicates the number of cells in the LSTM network. The 1st decay LR (Learning Rate) means the value set at the corresponding number of learning times reduced to 1/10 from the initial learning rate. The 2nd decay LR is a parameter that is set to prevent overfitting by reducing the learning rate reduced in the 1st decay LR to 1/10 again. Finally, the number of HN (Hidden Nodes) means the number of hidden nodes contained in a single cell. For example, in the first experiment presented in Table 6 below, there are 500 repetitions, 64 cells inside the LSTM network, and the initial learning rate value is 0.01. At this time, the learning rate is 0.001 at 250th, the 1st decay LR, and 0.0001 at 375th, the second decay LR. Further, the number of neurons inside the LSTM cells means that the number of neurons is set to 128 when creating the prediction model. A summary table (Table 6) of the results of the experiment is presented below. The sixth experiment showed a stroke prediction accuracy of 98.958% when it was predicted that 70% of the total data were randomly extracted, and the remaining 30% were not involved in learning. Consequently, the prediction of stroke based on the bio-signals of EMG showed stable predictions when setting the number of neurons inside LSTM Cells to twice or three times that of nUnit, with 2000 learning sessions and a learning rate of 0.001.

**Table 6.** Estimation accuracy of stroke disease by LSTM network structure (%).

| LSTM Number | Iteration | nUnit | Learning Rate | 1st Decay LR | 2nd Decay LR | Number of HN | Accuracy (%) | |
|---|---|---|---|---|---|---|---|---|
| | | | | | | | 70/30 | 80/20 |
| 1 | 500 | 64 | 0.01 | 250 | 375 | 128 | 92.70 | 92.66 |
| 2 | 500 | 256 | 0.001 | 250 | 375 | 768 | 94.77 | 94.80 |
| 3 | 1000 | 128 | 0.01 | 500 | 750 | 384 | 95.82 | 95.86 |
| 4 | 1000 | 256 | 0.001 | 500 | 750 | 768 | 96.44 | 96.48 |
| 5 | 2000 | 64 | 0.01 | 1000 | 1500 | 128 | 96.870 | 96.924 |
| 6 | 2000 | 128 | 0.001 | 1000 | 1500 | 384 | 98.958 | 98.952 |
| 7 | 3000 | 128 | 0.01 | 1500 | 2250 | 384 | 97.78 | 97.74 |
| 8 | 3000 | 256 | 0.001 | 1500 | 2250 | 768 | 98.56 | 98.62 |
| 9 | 5000 | 128 | 0.01 | 2500 | 3750 | 384 | 97.86 | 97.88 |
| 10 | 5000 | 256 | 0.001 | 2500 | 3750 | 768 | 98.74 | 98.74 |

Figure 6a below shows the accuracy of the prediction when the learning data is 70% and the test data is 30% of the sixth experiment described in Table 6. Figure 6b shows an EMG-based stroke disease prediction model that reduces errors and ensures optimal performance with repeated learning.

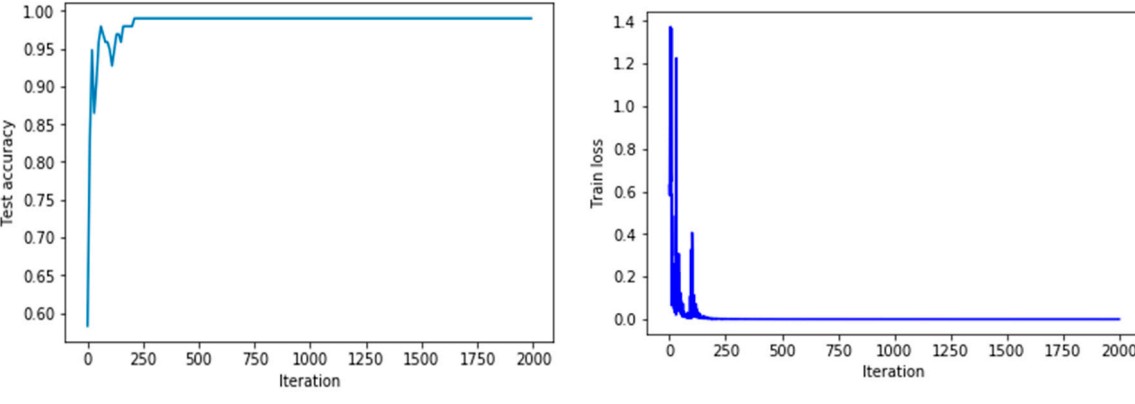

(**a**) Prediction accuracy with number of iterations      (**b**) Error rate with number of iterations

**Figure 6.** Trend of Prediction Accuracy and Error Rate with LSTM Learning. (**a**) Prediction accuracy with number of iterations, (**b**) Error rate with number of iterations.

### 4.4. Results and Discussions

In this paper, EMG bio-signals while walking were collected and the prediction accuracy of stroke diseases of 90.38% by Random Forest of machine learning and 98.958% by LSTM algorithm of deep learning were obtained. The system proposed in this study utilized real-time EMG data in four locations, left and right biceps femoris and gastrocnemius muscle, at 1500 Hz from EMG healthcare devices. The results of the analysis and prediction of stroke diseases based on these real-time bio-signals is a good approach as it is significant for the medical staff and hospitals to take a preemptive measure for early detection and prediction diagnosis of stroke diseases in patients in the danger group.

Since machine learning and deep learning are AI (artificial intelligence) methodologies, model generated should be comprehensively interpreted together by clinical diagnosis experience and knowledge of medical staff, electronic medical records, clinical data, emergency blood test and MRI (magnetic resonance imaging) information, etc. to ensure accurate stroke disease analysis and predictions. Therefore, it is desirable to combine stroke diseases prediction studies, which are analyzed by using medical knowledge and clinical experimental results of the medical team, rather than just using the AI-based model develop in this paper by itself. In addition, since the service proposed in this paper is the result of experimenting with EMG bio-signals only during walking, the prediction of multi-modal should also be considered by collecting multimodal bio-signals such as EEG (electroencephalogram) and ECG (electrocardiogram) as well. Finally, we hoped that research for early detection and prediction of strokes and chronic diseases based on bio-signals will be actively carried out by expanding into daily life activities such as sleep and driving. From this paper, EMG-based experimental results in this study are aimed at helping medical staff accurately predict and diagnose stroke diseases, and initial results showed that it is significant that AI methodologies and bio-signals collected can be used to help diagnose diseases only in everyday life.

## 5. Conclusions

We presented a system using the Random Forest algorithm of machine learning and LSTM of deep learning that can detect and predict stroke based on real-time EMG bio-signals. The proposed system can significantly minimize the social and economic losses from stroke by predicting stroke in real-time. In this paper, four points of EMG raw data were measured and collected in real-time from

healthcare devices. From this raw data, attributes were extracted and used with ML/DL models for stroke prediction accuracy and verification of this system. In addition, our experiment has shown that it is possible to detect and predict stroke symptoms with only bio-signals generated in everyday activities. This system overcomes the limitations of other systems by providing the probability of the occurrence of a disease in the next 10 years, or the degree of severity after the outbreak. This prediction model is significant as it can reduce misdiagnosis levels and be used to warn medical staff or hospitals to preemptively respond to the health needs of older people through early detection and prediction of stroke diseases. Additionally, the proposed Random Forest algorithm and LSTM-based stroke disease prediction model have the advantage of being able to be extended and applied as an early prediction model for diseases such as heart disease. In addition, we will develop a system that can analyze various type of data (diagnostic experience, medical knowledge, biometric signal analysis information, and EMR, etc.) in a scientific and comprehensive way so that it can directly help the diagnosis and decision of the medical doctors.

Future research tasks include measuring and collecting real-time bio-signals in driving or sleeping services as well as walking during everyday life, along with conducting research and development on the provision of more comprehensive forecasting services for stroke diseases.

**Author Contributions:** Conceptualization, J.Y. and C.-S.P.; methodology, J.Y., H.L., C.-S.P., S.P., S.-H.K. and C.M.B.H.; software, J.Y., C.M.B.H.; validation, J.Y., H.L., C.-S.P., S.P., S.-H.K. and C.M.B.H.; formal analysis, J.Y., H.L., C.-S.P., S.P., S.-H.K. and C.M.B.H.; investigation, J.Y. and C.-S.P.; resources, J.Y. and C.-S.P.; data curation, J.Y. and H.L.; writing—original draft preparation, J.Y., H.L., C.-S.P., S.P., S.-H.K. and C.M.B.H.; writing—review and editing, J.Y. and H.L.; visualization, J.Y.; supervision, J.Y., H.L. and C.-S.P.; project administration, C.-S.P.; funding acquisition, C.-S.P. All authors have read and agreed to the published version of the manuscript.

**Funding:** This work was supported by the National Research Council of Science & Technology (NST) grant by the Korea government (MSIP) (No. CRC-15-05-ETRI).

**Conflicts of Interest:** The authors declare no conflict of interest.

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
