# Peer review of "AI-Based Stroke Disease Prediction System Using Real-Time Electromyography Signals"

_applsci, doi:10.3390/app10196791_

Round 1

Reviewer 1 Report

In this paper the authors presented a system using the Random Forest algorithm of machine learning and LSTM of deep learning that can detect and predict stroke based on real-time EMG bio-signals.  This methodology can be considered as an low-cost alternative, that can obtain accurate stroke prediction and can potentially be used for other diseases.

However in my opinion as an invasive methodology it has  a minor chance to become the principal or primary investigation in stroke subjects.

This manuscript is well-prepared, informative wih scientific soundness.

Nevertheless, some style and English grammar should be improved.

Additionally, some issueses regarding stroke should be corrected:

Line 111- remove headache-  it s not typical syndrome of stroke and neural cells lost of function,

Line 114- large arteriosclerosis???- did You mean large artery atherosclerosis?

  • it is not a main  and the most common cause of stroke- the most common is cardioembolism- please correct it,

Line 118- hypoplasia is very rare cause of stroke and it should not be presented here, and is not related to blood coagulation disease- please remove it.

Line 125- subarachnoidal hemorrhage is not a subclinical hemorrhage- it always  lead to dramatic symptoms- please change it.

Author Response

Dear Reviewer.

I upload the response file.

Best regards,

Hansung Lee.

Reviewer 2 Report

This paper proposes the use of real-time electromyography data, obtained from everyday activities, to detect and predict stroke. The contribution is well presented with a solid overview of the relevant literature, and the authors explain their motivation with clarity. However, there are some areas where improvements are needed:

  1. The paper does not include a discussion section; this section is inalienable in a paper as it is where the results should be interpreted, what is the meaning of the research, possible implications, improvements, and future research in the area. In my opinion, a discussion section should be added.
  2. The results are interesting, but I feel that some important information has been overlooked. For example, there is no information about the report or information sent to the practitioners or the medical services, is this a number (%), an estimation, a sentence, or what? Has the practitioner the possibility of reviewing why the result is as it is? Or correct when the model is wrong? Another important aspect that has been not included in the paper is the user experience. The model relays in the real-time measurements for EMG bio-signal and for that some sensors should be placed in several parts of the body, mainly in the user’s legs, I feel that this can cause some discomfort in the users, so their opinion is key to envisage if this technique can be applied or not in a real scenario. Have the users measure somehow the user experience? Do you have any results in this regard?
  3. Thought the paper I can see how the technique could help to detect or anticipate a new stroke episode in a population that has suffered a previous one, but I cannot really see how this may be done in a “healthy” population.
  4. During the paper there are few issues with the references, for example in line 39 the reference 5 talks about the top 10 causes of death not about the rapid aging of the population, in the same way, in lines 53-56 it is used the same reference (number 5) to explain the leading causes of death, but this does not correspond with the information of the reference. Line 61 says various studies and clinical trials, but no reference is included. In line 147 the reference for Brott et al. is missed. In lines 173 and 177, it is said clinical studies… without the corresponding reference. Reference 38 is a newspaper entry, so maybe it can be reconsidered within a scientific paper. Line 298 says comprehensive studies but no reference or references are included.
  5. Line 348 says that at 1500Hz per second EMG, the variation in muscle movement is considered to be acceptable, some reference could be added to support this idea.
  6. The image of the measurement locations is included twice, in figure 3, and as a part of figure 1, consider to delete one of them.
  7. The introduction states the disadvantages of some prediction models, but these disadvantages are not explained (lines 83-84).

Author Response

(The authors gave the same response as above.)

Round 2

Reviewer 2 Report

The authors responded properly to all comments and requirements. Based on this I recommend the publication of the work.